# Immediate Effects of Light Mode and Dark Mode Features on Visual Fatigue in Tablet Users

**DOI:** 10.3390/ijerph22040609

**Published:** 2025-04-12

**Authors:** Praphatson Sengsoon, Roongnapa Intaruk

**Affiliations:** 1Department of Physical Therapy, School of Allied Health Sciences, Walailak University, Nakhon Si Thammarat 80160, Thailand; roongnapa.in@wu.ac.th; 2Movement Sciences and Exercise Research Center, Walailak University (MoveSE-WU), Nakhon Si Thammarat 80160, Thailand

**Keywords:** critical flicker frequency, dry eye, light and dark mode features, tablet user, visual fatigue

## Abstract

**Background:** Prolonged tablet use can cause visual fatigue, dry eye, and changes in critical flicker frequency, affecting visual comfort and performance. While the light and dark mode features aim to reduce eye strain, their immediate effects on these issues remain unclear. **Objective:** To compare the immediate effects of light and dark mode features on visual fatigue in tablet users. **Methods:** This experimental study involved 30 female tablet users. The participants were randomly assigned using a block randomization method to use both light and dark mode features. Visual fatigue, critical flicker frequency, and dry eye symptoms were measured before and after exposure to each mode. **Results:** No statistically significant difference in visual fatigue was observed between the two modes (*p* > 0.05). However, a statistically significant difference was found in critical flicker frequency (*p* < 0.05) and dry eye symptoms (*p* < 0.05) between the two modes. Furthermore, visual fatigue, critical flicker frequency, and dry eye symptoms significantly increased after tablet use in both modes (*p* < 0.05). **Conclusions:** Both light and dark mode features impact visual fatigue, critical flicker frequency, and dry eye symptoms. However, the dark mode may help reduce the risk of eye fatigue compared to the light mode. Further studies are recommended to explore the long-term effects and optimize screen settings for visual comfort.

## 1. Introduction

One important device that has evolved in today’s society, defined as a learning society that incorporates media and technology into education, is the tablet. According to a survey on internet usage behavior in Thailand for 2022, approximately 24.38% [1] of internet users accessed the internet through tablets for educational research. This highlights the significant role tablets play in enhancing learning efficiency. Moreover, with the increasing publication of e-books, tablets are being utilized to boost student motivation and positively impact learning outcomes while also facilitating self-directed learning among the general public, students, and university students [2]. However, despite these benefits, long-term tablet usage can lead to eye-related issues, such as myopia and eye strain [3,4].

Studies indicate that the prevalence of Computer Vision Syndrome (CVS) among computer users in Thailand is 81% [5], while a study in the United States found that 75% of users experienced eye strain [6]. Additionally, Pakistan showed the highest prevalence, with a 97% prevalence rate for CVS [7]. This research collectively points to a significant issue: most tablet users experience eye fatigue, which correlates with both gender [8] and age [9].

A study on eye strain assessment indicates that this condition arises from muscle tension caused by prolonged focus on objects. Women are more likely to suffer from eye strain, likely due to the smaller size of their eye muscles compared to men, resulting in greater fatigue during prolonged tasks that require intense eye muscle engagement [8]. Moreover, the prevalence of eye strain in computer users is twice as high in women compared to men [8]. Other studies also indicate that age has a statistically significant correlation with eye strain, showing that individuals over 40 years old are 2.39 times more likely to experience CVS compared to those under 40 years old [9].

The other cause of eye strain and myopia is the blue light emitted from tablet screens, which falls within the electromagnetic spectrum at wavelengths of 400–700 nanometers. This high-frequency, short-wavelength light can be harmful to the eyes, particularly the wavelengths ranging from 415–455 nanometers, which focus on the retina. Prolonged exposure to blue light can negatively impact the retina and surrounding eye muscles [10]. Currently, companies like Apple and Android are continuously developing effective technologies to address these issues. One such development is the ability to adjust screen display settings, including brightness levels, color tones, pixel density, font sizes, and more. Features such as light mode and dark mode have been introduced to alleviate eye strain. Light mode features black text on a white background, while dark mode displays white text on a black background. The research indicates that the dark mode can reduce eye strain compared to the light mode [11].

Despite advancements in display technologies, eye strain remains a prevalent issue among tablet users. A study on the effects of polarity and luminance contrast on visual performance and display quality found that while polarity had no impact, luminance contrast significantly influenced both visual acuity and perceived display quality. These results suggest that optimizing luminance contrast could help improve display quality and reduce eye strain, providing valuable insights for improving tablet display technology and enhancing user comfort [12]. Symptoms of eye strain include discomfort around the eyes, fatigue, blinking, and dryness. However, while existing studies have explored the effects of screen display settings on eye strain, there remains a gap in understanding how different screen modes—specifically the light mode and dark mode—affect not only eye strain but also dryness and visual fatigue in users. Most participants work in brightly lit environments with high-brightness screens and negative polarity settings. Such conditions create high contrasts in luminance, especially between the screen and surfaces within the field of vision. The results indicate visual fatigue during extended computer use, which, in most cases, leads to difficulty in continuing the activity and changes in their usual working behavior [13]. Previous research has focused primarily on general symptoms or isolated factors such as brightness, but few studies comprehensively compare the immediate effects of these screen modes on eye strain and discomfort.

This research aims to fill this gap by systematically comparing eye strain and dryness experienced by tablet users while using light mode versus dark mode. The goal is to assess the immediate differences between these two display modes in terms of visual fatigue, dryness, and overall comfort, contributing valuable insights into the optimal usage of tablet screens for prolonged periods.

## 2. Materials and Methods

This study compared the immediate effects of light mode and dark mode on eye strain among tablet users by using an experimental design (crossover study) using a pre-test–post-test design. The eligible participants were general tablet users, females aged between 18 and 25 years, with a body mass index (BMI) ranging from 18.5 to 22.9 kg/m^2^. In addition, the participants were those who used tablets regularly for at least five days a week, for at least two hours without interruptions, for at least 1 year of experience using tablets, and also had visual acuity of 20/20. However, they were excluded if they had eye disorders such as color blindness, cataracts, myopia, hyperopia, astigmatism, or strabismus, as well as neurological conditions affecting tablet use (e.g., vertebrobasilar insufficiency). Participants who consumed alcohol or caffeinated beverages within 24 h before testing and those experiencing mental health issues or stress on the testing day with scores greater than 25 on the Department of Mental Health’s assessment (SPST-20) were also excluded.

### 2.1. Instruments

The iPad (7th to 9th Generation), Copyright © 2023 Apple Inc., is manufactured by Foxconn Technology Group, which is located in China.The assessment of visual fatigue is divided into two types as follows:

The visual fatigue questionnaire (asthenopia questionnaire), which had excellent reliability (ICC_s_ = 0.899) [14], asks participants to describe their level of visual fatigue. This questionnaire consists of 10 items related to symptoms of visual fatigue, including eye fatigue, eye itching, eye irritation, watery eyes, dry eyes, eye pain, burning eyes, blurred vision, difficulty in focusing, and general discomfort in the eyes. The participants rate each symptom on a scale from 1 to 6, with the following scoring criteria: 1 indicates no symptoms, 2 indicates mild symptoms, 3 indicates minor symptoms, 4 indicates moderate symptoms, 5 indicates severe symptoms, and 6 indicates very severe symptoms, respectively. The total score ranges from 0 to 60, where a higher score indicates severe visual fatigue. The participants completed the visual fatigue questionnaire both before and after using the tablet in both bright and dark modes.

Critical flicker frequency measurement, which measures the CFF, is based on the principle of flicker perception and has been reported to have excellent reliability (ICC_s_ = 0.950) [15]. It involves an electronic device that alternates between emitting light and turning it off at increasing frequencies, starting from 20 Hz. The participants were instructed to observe a flashing color strip, and the frequency was gradually increased until they could no longer perceive the flicker. The measurement is recorded in cycles per second (Hertz), and each participant undergoes three trials, with each trial lasting approximately 20 s. The values were documented, allowing for the assessment of visual fatigue before and after reading on the tablet in both modes. The participants who were not experiencing eye strain could detect flickers at frequencies of 35–40 Hz. When eye strain occurs, the test yields a critical flicker frequency of 40 Hz or higher [16].

The dry eye questionnaire was used to assess dryness of the eyes and consists of 14 questions, including symptoms such as eye fatigue, discharge, the sensation of having a foreign body in the eye, heaviness of the eyes, dryness, discomfort in the eyes, excessive tearing, blurred vision, itchiness, sensitivity to light, redness, pain in the eyes, difficulty seeing (at close distances), and difficulty seeing (at far distances). The scoring criteria are as follows: 1 indicates no symptoms, 2 indicates minimal symptoms, 3 indicates mild symptoms, 4 indicates moderate symptoms, 5 indicates severe symptoms, and 6 indicates the most severe symptoms. It has been reported as an excellent content validity index (CVI = 0.900) [17]. The participants were assessed for eye dryness using the dry eye questionnaire both before and after using the tablet in both light and dark modes.

### 2.2. Procedures

The sample size was determined using the G*Power version 3.1.9.7 [18]. The researcher then used a questionnaire to screen all participants based on the inclusion and exclusion criteria before they participated in the study. The participants received an explanation regarding the research objectives, the research procedures, and the benefits of participating and then signed a consent form to participate in the research. Following this, the researcher assessed visual acuity using the Snellen chart to ensure that the participants’ visual ability met the standard of 20/20. The workstation was ergonomically designed, featuring a standard-height table set at 75 cm, with the screen positioned perpendicular to the participant’s line of sight when seated upright. Additionally, the adjustable chair allowed for height adjustments, and the screen brightness was set to 100 percent to maintain consistency across participants. The lighting conditions in the experiment were artificial, with the environment set up to ensure consistent and controlled illumination throughout the test. The room’s lighting was maintained at a level between 300–500 lux, which is considered optimal for this type of task, and the brightness of the tablet screen was also carefully controlled to maintain consistent illumination.

Participants were randomly assigned to groups through block randomization to select between Program 1 and Program 2, which determined the sequence for using the light and dark modes. For example, if a participant was assigned to Program 1, they would begin by engaging in activities where, in the first hour, they would spend 30 min playing a game and 30 min reading a novel on the tablet screen in light mode. Afterward, the participants would take a 30 min break, or they could extend the break until they no longer experienced any symptoms of eye strain, such as discomfort or dryness. Following the break, they would resume activities, including playing games and reading, using the tablet screen in dark mode to assess its impact on eye strain and visual comfort.

Before the test began, the participants familiarized themselves with the tablet device for 5 min. Prior to testing with the tablets, a critical flicker frequency machine was used to establish baseline normalization of the settings, ensuring consistency across all participants. This step helped to control key variables for the testing procedure. Following this, the participants underwent an eye strain assessment, completed the dry eye questionnaire, and were tested for visual fatigue using the critical flicker frequency machine. This assessment was conducted before the participants used the tablet screen in light mode. Additionally, the participants filled out the asthenopia questionnaire and the dry eye questionnaire, with their scores falling within the normal range. They then engaged in activities involving playing a game for 30 min and reading a novel for 30 min on the tablet under Mode no.1 for a total of 1 h [19], with all participants required to complete a medium-level game. Upon completion of the testing in Mode no.1, participants underwent a comprehension test (post-test) regarding the content of the novel they read, consisting of three questions, which they needed to answer correctly with a score of over 80 percent. After successfully passing the test, the participants repeated the assessments for eye strain, completed the dry eye questionnaire, and were tested for eye fatigue again using the critical flicker frequency machine. They then took a break for at least 30 min [19] or until they no longer experienced any symptoms of eye strain, verified through assessments of eye strain and dryness. The scores obtained from the questionnaires were required to indicate no signs of eye strain or dryness.

At the time of starting the subsequent testing phase, the participants completed questionnaires to assess eye strain and a dry eye questionnaire and underwent testing for eye fatigue using a critical flicker frequency machine before the assessment in Mode no.2. The participants filled out the asthenopia questionnaire and the dry eye questionnaire, scoring within the normal range. They then engaged in activities that included playing a game for 30 min and reading a novel for 30 min on the tablet in Mode no. 2 for a total of 1 h, with all participants required to complete a medium-level game. Upon completion of the testing in Mode no. 2, the participants took a comprehension test (post-test) regarding the content of the novel they read, consisting of three questions, which they needed to answer correctly with a score of over 80 percent. After successfully passing the test, the participants repeated the assessments for eye strain, completed the dry eye questionnaire, and were tested for eye fatigue again using the critical flicker frequency machine following the use of the tablet in this mode. The collected data were then analyzed for research findings, summarized, and subjected to statistical analysis. The details of the research process flowchart are shown in Figure 1.

## 3. Statistical Analysis

Statistical analysis was conducted using SPSS (version 27), with a significance level set at *p* < 0.05. In addition, visual fatigue, critical flicker frequency, and dry eye were analyzed using the Wilcoxon signed-rank test to compare the differences in these variables when using the tablet screen in both light mode and dark mode, as well as before and after using the tablet in both modes.

## 4. Results

This study aimed to compare the immediate effects of the light mode and dark mode on visual fatigue in tablet users. This research investigated the outcomes related to levels of visual fatigue, critical flicker frequency, and degrees of dry eye in a sample of female participants who used a tablet continuously for 1 h.

### 4.1. Demographic Characteristics

According to the statistical analysis, the demographic characteristics of the 30 female participants comprised an average age of 21.20 ± 1.16 years, weight of 52.82 ± 4.67 kg, height of 160.03 ± 5.26 cm, and a body mass index (BMI) of 20.64 ± 1.25 kg/m^2^. Moreover, the average duration of tablet use was 11.63 ± 1.71 h per day, and the average experience with tablet use was 4.40 ± 0.77 years.

### 4.2. Differences in Visual Fatigue When Using the Tablet Screen in Light Mode and Dark Mode, as Well as Before and After Using the Tablet Screen in Both Modes

The statistical analysis to determine the differences in visual fatigue before using the tablet screen in light mode and dark mode revealed no significant differences (*p* > 0.05). Similarly, no significant differences in visual fatigue were found when comparing the use of the tablet screen in both modes (*p* > 0.05). However, when comparing visual fatigue before and after using the tablet screen in light mode, significant differences were observed (*p* < 0.001). Similarly, significant differences were also observed in dark mode (*p* < 0.001), as presented in Table 1.

Although the differences between the light mode and dark mode were not statistically significant, it is important to consider the practical implications of these findings. While prolonged use of the tablet screen in either mode does lead to visual fatigue, users may still experience meaningful discomfort, even without statistical differences between the modes. Therefore, despite the lack of significant statistical differences, the results suggest that prolonged tablet use can lead to visual fatigue in both modes.

### 4.3. Differences in Critical Flicker Frequency When Using the Tablet Screen in Light Mode and Dark Mode, as Well as Before and After Using the Tablet Screen in Both Modes

The statistical analysis to determine the differences in CFF before using the tablet screen in light mode and dark mode revealed no significant differences (*p* > 0.05). However, when comparing using the tablet screen in light mode and dark mode, significant differences were found (*p* < 0.05). Additionally, when comparing CFF before and after using the tablet screen in both modes, significant differences were observed (*p* < 0.001), as presented in Table 2.

While these statistical findings indicate that the display mode and usage duration have a significant impact on CFF, the practical significance of these results should also be considered. Although the changes in CFF between the light mode and dark mode are statistically significant, the real-world implications for users may depend on the intensity and duration of tablet use.

### 4.4. Differences in Dry Eye When Using the Tablet Screen in Light Mode and Dark Mode, as Well as Before and After Using the Tablet Screen in Both Modes

The statistical analysis to determine the differences in dry eye before using the tablet screen in light mode and dark mode revealed no significant differences (*p* > 0.05). However, when comparing dry eye between using the tablet screen in both modes, significant differences were found (*p* < 0.05). Additionally, when comparing before and after using the tablet screen in both the light and dark modes, significant differences were observed (*p* < 0.001), as presented in Table 3.

While the statistical results show significant differences between modes and before/after usage, it is also important to consider the practical implications of these findings. The significant difference in dry eye symptoms between the two modes suggests that users may experience less discomfort in dark mode, particularly after prolonged use of the tablet. This could be important for users who are particularly sensitive to dryness or those who use tablets for long periods in environments with low lighting.

## 5. Discussion

### 5.1. Comparing the Variations in Visual Fatigue Before and After Using the Tablet Screen in Both Light and Dark Settings, as Well as When Using the Screen in Both Modes

The results of this study indicate that when comparing visual fatigue before and after using the tablet screen in light mode and dark mode, there is a statistically significant difference. Prolonged tablet use increases the risk of visual fatigue, which occurs due to the continuous contraction of the eye muscles over an extended period. This leads to abnormal mechanisms in visual function, as using a tablet involves focusing on close-up images. Consequently, the eyes must adjust to achieve optimal image clarity, with the crystalline lens in the eye responsible for focusing light onto the retina. Additionally, the ciliary muscle works to accommodate the lens for closer vision, resulting in contraction to maintain sharpness. The medial rectus muscles also contract to facilitate convergence of both eyes. Therefore, prolonged use of a tablet screen adversely affects the ability to accommodate and converge, ultimately leading to visual fatigue [20].

Additionally, visual fatigue can result from interference by light emitted from the tablet screen, particularly blue light with wavelengths ranging from 415 to 455 nanometers, which directly contributes to visual fatigue in tablet users. However, this study controlled the ambient light intensity in the workspace to remain between 450 and 500 lux, a level that minimizes eye strain. If the light intensity exceeds 500 lux, it can lead to excessive reflections from the tablet screen into the eyes of the user, necessitating an increased effort to focus [10]. Such reflections can cause discomfort and contribute to visual fatigue, as well as a decrease in visual performance [10]. Conversely, if the ambient light intensity is below 300 lux, which is considered too low, the eye muscles must work harder to dilate the pupils for clearer vision, leading to further visual fatigue [20].

Moreover, the findings of this study indicate that when comparing visual fatigue during the use of a tablet screen in the light and dark modes, no statistically significant differences were observed. However, there is a tendency for the dark mode to contribute more to visual fatigue than the light mode. The dilation of the pupil in low-light conditions, such as dark mode, allows more light to enter the eye but also leads to increased aberrations, thereby reducing image sharpness and enhancing spherical aberration [21]. Additionally, the study by Pedersen et al. (2020) found no significant differences between the dark mode and light mode among participants from Oslo Metropolitan University and Kristiania University College; however, participants preferred the dark mode [22]. Similarly, the study by Lu Cheng et al. (2024) noted that the brightness of the tablet screen did not significantly affect visual fatigue, as participants using an iPad Air 2 at brightness levels ranging from 0 to 50 percent and 0 to 100 percent showed no statistically significant differences in visual fatigue [23].

The findings of this study were consistent with the previous research outcomes, as evidenced by the eye fatigue questionnaire designed to assess symptoms of visual fatigue in participants. This assessment was conducted during tablet use, as well as before and after using the tablet in both the light and dark modes. The results indicated that the participants did not experience significant differences in visual fatigue between the two modes. Furthermore, the nature of the tasks assigned to the participants did not sufficiently provoke the severity of eye fatigue, leading to the conclusion that the participants were unable to perceive a notable difference in visual fatigue between the two modes.

### 5.2. Comparing the Variations in Critical Flicker Frequency Before and After Using the Tablet Screen in Both Light and Dark Settings, as Well as When Using the Screen in Both Modes

Generally, the instrument used to measure CFF establishes that normal values for adults fall within the range of 35 to 40 Hz [24]. Additionally, age is a significant factor affecting CFF, as the frequency tends to decrease with advancing age. This decline is attributed to the loss of flexibility in the eye’s lens, which impairs the function of the ciliary muscles responsible for adjusting the eye’s power for near vision. Consequently, this makes it more challenging to adjust the focal length for clarity, leading to a reduced perception of flicker frequency [25].

Furthermore, the results of this study indicated a significant statistical difference when comparing CFF between the use of tablets in light mode and dark mode. The data showed a tendency for higher CFF values in light mode compared to dark mode. This difference can be attributed to the visual mechanism involved when viewing a tablet screen, which involves the refraction of light through the cornea and lens. These structures adjust the focal length to achieve clarity on the retina. Prolonged focus on text displayed on a tablet screen requires the eyes to continuously adjust their focus, causing the ciliary muscles to contract automatically to maintain clear vision. Extended periods of muscle exertion can lead to visual fatigue, potentially resulting in increased frequency measurements when assessing the eye’s perception of flicker [26].

Moreover, the results of the present study revealed a statistically significant difference in CFF when comparing the use of tablets in light mode versus dark mode. The data indicated a trend toward higher CFF values in light mode compared to dark mode, suggesting a correlation with visual fatigue.

### 5.3. Comparing the Variations in Dry Eye Before and After Using the Tablet Screen in Both Light and Dark Settings, as Well as When Using the Screen in Both Modes

The results of this study demonstrate that when comparing dry eye symptoms before and after using the tablet in both light and dark modes, there is a statistically significant difference. Prolonged use of the tablet screen necessitates near-vision focusing, which reduces the frequency of eye blinking, leading to an increased evaporation rate of tears and resulting in dry eyes. This also causes abnormalities in the tear film and dysfunctional surface properties of the eyes [26]. Additionally, continuous use over an extended period affects the ciliary body, which is responsible for tear production and is also associated with adjusting focus for different distances. Prolonged activity can lead to fatigue and decreased functionality, resulting in blurred vision and diminished capacity to adjust focus at varying distances. These symptoms are correlated with dry eye conditions [27]. The study investigated the relationship between computer use and dry eye symptoms. It was found that extended computer use reduces blink rates, leading to dry eye symptoms, which is one of the manifestations of visual fatigue and ultimately results in decreased visual performance [28].

Moreover, previous studies have reported that most computer users experience dry eye symptoms, primarily caused by two factors, including decreased tear production from the lacrimal glands and increased tear evaporation. A significant contributing factor is computer screen usage [28,29]. Consequently, computer use can lead to a reduction in blink rate, resulting in dry eyes. Under normal conditions, the blink rate averages between 9 to 17 times per minute [30]. Blinking is facilitated by the action of the orbicularis oculi muscle, which stimulates the lacrimal glands to produce tears that coat the eye’s surface, keeping it moist and preventing dryness [19,27]. Tears can be classified into the three following layers: The outermost lipid layer, which protects against tear evaporation; the middle aqueous layer, which helps eliminate foreign particles and pathogens; and the innermost mucin layer, which ensures even distribution of tears across the cornea. Each of these three layers plays a crucial role in protecting and hydrating the eyes. If any one of these layers becomes dysfunctional, it can lead to dry eye conditions [31,32].

Additionally, computer use can lead to incomplete blinking, where the upper and lower eyelids do not close completely, resulting in increased tear evaporation, which can also cause dry eyes. Furthermore, using a computer involves a horizontal gaze, which creates a wider gap between the upper and lower eyelids compared to typical reading activities. This increased surface area of the eye results in a higher rate of tear film evaporation, leading to dry eye symptoms [33,34].

Furthermore, the present study showed that when comparing dry eye symptoms between the use of the light mode and dark mode, there is a tendency for a difference in the level of dry eye fatigue. The higher screen brightness allows more light to enter the eyes, increasing the likelihood of dry eyes, which may contribute to visual fatigue and other related issues. Conversely, lower screen brightness reduces the amount of light entering the eyes, thereby decreasing the risk of dry eyes, which corresponds with the dark mode [35].

## 6. Conclusions

This study investigated the immediate effects of light mode and dark mode on visual fatigue, critical flicker frequency (CFF), and dry eye symptoms in a group of 30 female participants who used a tablet continuously for one hour. This study fills an important gap in the literature by exploring the impact of light mode and dark mode on visual fatigue and eye health during prolonged tablet use. While the effect of screen settings on visual discomfort has been studied in various contexts, few studies have specifically compared light mode and dark mode in terms of their influence on symptoms like visual fatigue and dry eye. Our findings contribute valuable insights into how these modes can affect user comfort and well-being.

Based on the results, we recommend that users who experience visual fatigue or dry eye symptoms during extended tablet use consider switching to dark mode, as it may help reduce the level of discomfort. Additionally, users should take regular breaks, adjust the screen’s brightness to a comfortable level, and consider using blue light filters to further alleviate strain on the eyes. These simple adjustments can significantly improve the user’s experience, especially for those who rely on digital devices for prolonged periods.

This study underscores the importance of screen settings in minimizing visual strain, offering practical guidance for everyday users and digital device developers. By addressing a gap in our understanding of how the light and dark modes influence eye health, this research highlights the need for further exploration into optimizing screen settings for better user comfort.

## 7. Limitations of the Study

### 7.1. Visual Acuity of Participants

The participants in this study had normal visual acuity, measured using the Snellen chart with a requirement of 20/20 vision. This was done to control for factors that could contribute to visual fatigue, such as individuals with myopia, who may experience greater visual fatigue when using a tablet for extended periods compared to those with normal vision. This poses a limitation in generalizing the study results to populations with abnormal visual acuity.

### 7.2. Gender of Participants

The participants in this study were all female, which limits the generalizability of the results to male populations.

### 7.3. Age Groups and Lighting Conditions

Another limitation of this study is that the sample consisted of participants within a specific age range, which may affect the generalizability of the findings to other age groups. Additionally, this study was conducted under standard lighting conditions, which may not reflect the diverse lighting environments in which people typically use tablets. Ambient lighting can influence visual fatigue, particularly in bright or dimly lit conditions.

## 8. Future Studies

Future studies will include investigating eye strain in populations with visual impairments or abnormal levels of vision, such as individuals with myopia or hyperopia, when using tablet screens in light or dark modes. Moreover, they will include examining the long-term effects and residual impacts of eye strain from using tablet screens at varying modes. This would provide a deeper understanding of the sustained effects on eye health for users who engage in extended tablet use.

## Figures and Tables

**Figure 1 ijerph-22-00609-f001:**
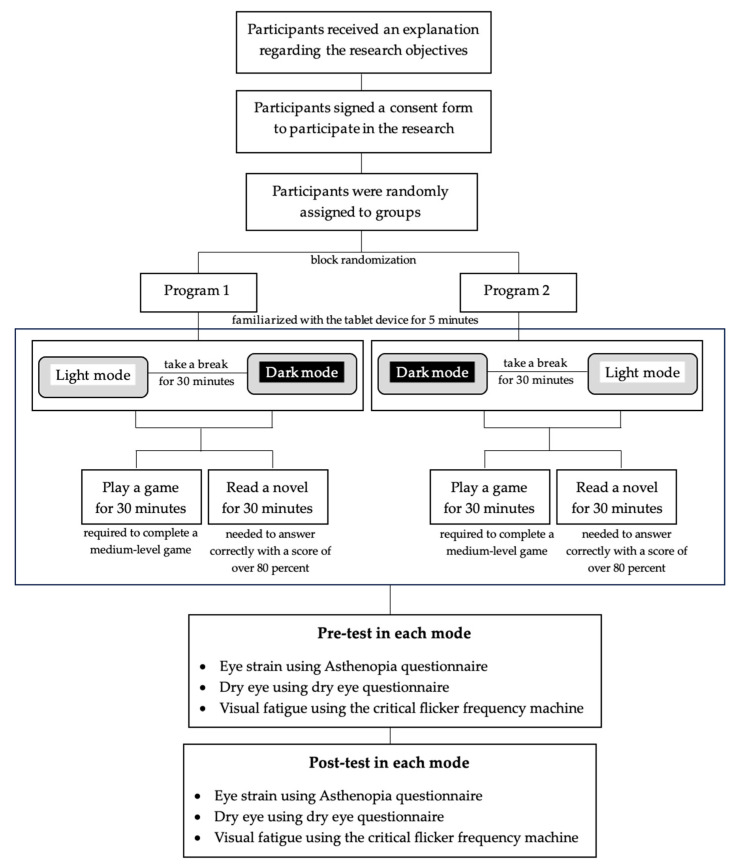
The research process flowchart.

**Table 1 ijerph-22-00609-t001:** Differences in visual fatigue when using the tablet screen in light mode and dark mode, as well as before and after using the tablet screen in both modes.

Mode	Visual Fatigue (Pre) (Score)			*p*-Value (Pre)	Visual Fatigue (Post) (Score)			Visual Fatigue Value (Different Changes) (Score)	*p*-Value (Pre-Post)	*p*-Value (Between Mode)
	Mean ± SD	Min	Max		Mean ± SD	Min	Max			
Tablet use with light mode	11.47 ± 1.676	10	17	*p* > 0.05	18.37 ± 6.955	10	34	7.067	*p* < 0.001	0.305
Tablet use with dark mode	12.40 ± 2.872	10	22		18.87 ± 7.011	10	38	6.267	*p* < 0.001	

**Table 2 ijerph-22-00609-t002:** Differences in critical flicker frequency when using the tablet screen in light mode and dark mode, as well as before and after using the tablet screen in both modes.

Mode	CFF Values (Pre) (Hz)			*p*-Value (Pre)	CFF Values (Post) (Hz)			CFF Values (Different Changes) (Hz)	*p*-Value (Pre-Post)	*p*-Value (Between Mode)
	Mean ± SD	Min	Max		Mean ± SD	Min	Max			
Tablet use with light mode	37.29 ± 1.071	33.87	38.6	*p* > 0.05	41.35 ± 1.548	39.25	46.1	4.042	*p* < 0.001	0.001
Tablet use with dark mode	37.43 ± 1.018	34.8	39.1		40.59 ± 1.854	36.65	45.45	3.162	*p* < 0.001	

**Table 3 ijerph-22-00609-t003:** Differences in dry eye when using the tablet screen in light mode and dark mode, as well as before and after using the tablet screen in both modes.

Mode	Dry Eye Values (Pre) (Score)			*p*-Value (Pre)	Dry Eye Values (Post) (Score)			Dry Eye Values (Different Changes) (Score)	*p*-Value (Pre-Post)	*p*-Value (Between Mode)
	Mean ± SD	Min	Max		Mean ± SD	Min	Max			
Tablet use with Light mode	16.93 ± 2.935	14	25	*p* > 0.05	24.40 ± 7.686	15	42	7.600	*p* < 0.001	0.048
Tablet use with Dark mode	16.33 ± 2.294	14	21		22.73 ± 6.373	14	36	6.400	*p* < 0.001	

## Data Availability

The data is unavailable due to privacy or ethical restrictions.

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
