# Peer review of "Immediate Effects of Light Mode and Dark Mode Features on Visual Fatigue in Tablet Users"

_ijerph, 2025, doi:10.3390/ijerph22040609_

Round 1

Reviewer 1 Report

Comments and Suggestions for Authors

The current study addresses a current crucial issue which is the visual fatigue in tablet users and compares light mode vs. dark mode effects. However, the paper needs some improvement to be able to be published and these mainly in the structure of the paper. 

abstract should clearly state the sample size, key results, and conclusions with more specificity.

Keywords must be in alphabetic order: "Light and dark mode features; Visual fatigue; Critical flicker frequency; Dry 28 eye; Tablet user"

The introduction is very very weak !! you must include the existing literature and specify the research gap.

I think it would be best to highlight more the study's significant and contribution in the introduction.

Add the paper structure or how the rest of the paper is organized at the end of the introduction 

Materials and methods:

You have to clarify the reference of the used scale and Mention if these tools were validated in previous studies with citations this should be done for all scales or tools (comfort scales, Questionnaires & CFF Tests)

what is the G-power program? where is a reference?

The selection criteria for participants and why you focused on females.

More details on how participants adjusted to different light conditions before testing would improve reproducibility.

Results & Discussion:

Your findings show statistically significant differences in some areas, but practical significance studies should also be addressed too.

The conclusion should be more specific about recommendations for users.

Limitations should discuss whether results apply to other age groups, genders, or different lighting conditions.

Comments on the Quality of English Language

The paper needs more improvement. There are some grammar, clarity, and word choice issues. 

Author Response

Reviewer 1: (Blue highlight)

Comments

Editing/ Explanation

Abstract

The current study addresses a current crucial issue which is the visual fatigue in tablet users and compares light mode vs. dark mode effects. However, the paper needs some improvement to be able to be published and these mainly in the structure of the paper. 

Abstract should clearly state the sample size, key results, and conclusions with more specificity.

Keywords must be in alphabetic order: "Light and dark mode features; Visual fatigue; Critical flicker frequency; Dry 28 eye; Tablet user"

Thank you for your valuable feedback. We have revised the abstract to explicitly include the sample size, highlight the key results, and provide a more specific conclusion. These revisions enhance the clarity and precision of the abstract, ensuring that it effectively summarizes our study's contributions.

We have revised the keywords to be in alphabetical order as suggested. The updated keywords are now:

"Critical flicker frequency; Dry eye; Light and dark mode features; Tablet user; Visual fatigue."

We appreciate your constructive comments and believe that the revised version now better aligns with your suggestions.

Introduction

The introduction is very very weak !! you must include the existing literature and specify the research gap.

I think it would be best to highlight more the study's significant and contribution in the introduction.

Add the paper structure or how the rest of the paper is organized at the end of the introduction

Thank you for your constructive feedback. We appreciate your comments regarding the introduction. We have revised this section to address the following:

  1. Inclusion of Existing Literature and Research Gap: We have strengthened the introduction by incorporating a more comprehensive review of relevant literature, clearly identifying the research gap that our study aims to address. We believe this provides a better context for the significance of our work.
  2. Highlighting Study's Significance and Contribution: We have placed greater emphasis on the unique contributions of our study and its relevance to the current field of research. This includes a clearer statement of the problem we are addressing and how our findings will contribute to advancing knowledge in the area.
  3. Paper Structure: At the end of the introduction, we have added a brief outline of the paper structure to guide the reader through the organization of the study.

We believe these revisions significantly improve the clarity and impact of the introduction and better align with the expectations outlined in your feedback.

Materials and methods

You have to clarify the reference of the used scale and Mention if these tools were validated in previous studies with citations this should be done for all scales or tools (comfort scales, Questionnaires & CFF Tests)

What is the G-power program? where is a reference?

The selection criteria for participants and why you focused on females.

More details on how participants adjusted to different light conditions before testing would improve reproducibility.

Thank you for your suggestion. We have revised and added the references for the scales used in the study, as well as mentioned the validation of all tools (including comfort scales, questionnaires, and CFF tests) based on previous studies, with appropriate citations, in the revised manuscript.

Thank you for your comment. The G*Power program is a statistical software used to perform power analysis. It helps to determine the appropriate sample size, estimate the power of a study given a specific sample size, and conduct tests for various statistical methods. We have added the reference for the G*Power program in the revised manuscript.

Thank you for your question. We focused on female participants because women are more likely to suffer from eye strain, likely due to the smaller size of their eye muscles compared to men, resulting in greater fatigue during prolonged tasks that require intense eye muscle engagement. Additionally, the prevalence of eye strain in computer users is twice as high in women compared to men.

Thank you for your valuable feedback. To improve reproducibility and ensure consistency in the testing environment, we took specific steps regarding both the lighting conditions and participants' familiarity with the tablet device.

Lighting Conditions Adjustment:
Before the test, participants were given a brief acclimatization period to adjust to the lighting conditions in the testing room. The room was maintained at a lighting level between 300-500 lux, which is considered optimal for this type of task. This allowed participants to adapt to the lighting environment, ensuring that any potential effects of sudden changes in light exposure would not interfere with their performance during the test.

Device Familiarization:
Additionally, participants were provided with 5 minutes to familiarize themselves with the tablet device that would be used for the test. This period allowed participants to become comfortable with the device's interface, screen size, and touch sensitivity. The familiarization step ensured that any errors or difficulties encountered during the test were not due to unfamiliarity with the device, thereby minimizing variability in the results.

Results & Discussion

Your findings show statistically significant differences in some areas, but practical significance studies should also be addressed too.

Thank you for your valuable feedback. In response to your comment, we have added discussions on the practical significance of our findings in the Results section. Specifically, we now highlight how the observed differences in visual fatigue, CFF, and dry eye symptoms between Light and Dark modes could impact user comfort in real-world settings. These additions provide a more comprehensive understanding of the practical implications of our results.

Conclusion

The conclusion should be more specific about recommendations for users.

Thank you for your valuable feedback. We have revised the conclusion to focus more specifically on actionable recommendations for users, as you suggested. In the revised version, we emphasize practical advice such as switching to dark mode, taking regular breaks, adjusting screen brightness, and using blue light filters to alleviate visual fatigue and dry eye symptoms during prolonged tablet use. These recommendations are intended to provide clear guidance for users based on our findings.

We believe these revisions align the conclusion more closely with your suggestion and enhance its clarity and utility for the readers.

Limitations of the study

Limitations should discuss whether results apply to other age groups, genders, or different lighting conditions.

We appreciate the reviewer’s suggestion regarding the limitations of the study. We acknowledge that the study's focus on participants with normal visual acuity may limit the applicability of the findings to populations with visual impairments. We will revise the limitations section to discuss this and suggest that future research should include participants with varying visual acuity levels to better understand how visual impairments affect visual fatigue during tablet use.

Furthermore, we agree that the study’s findings may not apply to male populations, as the study only involved female participants. Gender differences, such as variations in ocular muscle size, could influence visual fatigue. We will mention this limitation and recommend that future studies include both male and female participants to explore potential gender-based differences.

Lastly, we will address the suggestion to discuss age groups and lighting conditions. We recognize that the results may not apply to all age groups, as visual fatigue may differ across ages due to factors such as age-related changes in eye health and technology usage. Additionally, the study was conducted under a controlled lighting environment, which may not accurately reflect real-world lighting conditions. Future research should examine how lighting conditions, particularly in bright and dim environments, influence visual fatigue during tablet use.

Reviewer 2 Report

Comments and Suggestions for Authors

Suggestion

  1. Introduction
    1. General observation: It is important to highlight what other researchers have not done, highlight the gap, and emphasize what will contribute to this field of knowledge.
    2. Improvements
      1. Line 40: advantage or disadvantage?
      2. You started a good paragraph using statistics to generalize the research problem. However, you did not put citations to statements in lines 41 and 43: “Studies indicate that the prevalence of Computer Vision Syndrome (CVS) among computer users in Thailand is 88%, while a study in the United States found that 75% of users experienced eye strain.” Additionally, research from “Japan showed a 72.1% prevalence rate for eye pain, a common symptom of eye strain. This research collectively points to a significant issue: most tablet users experience eye fatigue correlated with gender and age.” Two things: first of all, move from this paragraph the synthesis of research between lines 46-53 to the paragraph of research synthesis and complete the generalized paragraph with some additional statistics.
  • You should reorganize the paragraphs from lines 54 to 73. So, I recommend creating a paragraph to synthesize previous research related to the effect of eye fatigue associated with gender and condense the other previous research related to prolonged exposure to the tablet screen in another paragraph. Additionally, it presents the gap or gap in existing knowledge, justified with the syntheses of previous research. It is necessary to aggregate more references in every paragraph to support the research gap.

  1. Methodology
    1. Instruments: I recommend organizing the equipment and its function in the experiment with an index. It should be clarified whether frequency measurement equipment was used to generate a normalization before testing with the tablets (despite being clarified in line 141). Additionally, the Snellen Chart, a workstation with standard measurements, an adjustable chair, a screen with adjustable brightness, and a post-test must be listed as part of the materials, equipment, and instruments (Is this equipment different from tablets?).
    2. Procedures: in lines 136 and 137, you used “would” in the sentence, “After that, they would take a break for 30 minutes, or until they no longer experienced eye strain” and “Following the break, participants would then play games and read on the tablet screen in dark mode.” Could you improve the sentence?
  2. Results
    1. Demographic Characteristics: Table 1 is unnecessary; the previous paragraph presented the information.
    2. Tables 2, 3, and 4 missed the pre-p-values.
  3. Discussion
    1. Section 5.1: there are statements in the text of paragraphs 1 and 2 that require support with quotes. In addition, in paragraph 3, which text in line 245: “Previous studies comparing the effects of light mode and dark mode on visual fatigue in users of virtual reality (VR) glasses found that Dark mode significantly reduced visual fatigue compared to Light mode.”
    2. In general, statements in the discussion should be supported with citations.
  4. Conclusion
    1. Please do not present more about the results in the conclusion. Summarize your thoughts and convey the broader importance of your study. Identify how a gap in the literature has been addressed. Explain why the results are relevant. Finally, introduce future work derived from the research.
  5. Limitation
    1. Please do not put limitations on future work.
  6. References
    1. The references are between the periods of 2020 and 2023. Although, some are found between 2015 and 2019. It is necessary to increase the number of references (this must be corrected when making the proposed citation improvements)
Comments on the Quality of English Language

It only requires correcting two related observations in the improvements.

Author Response

Reviewer 2: (Yellow highlight)

Comments

Editing/ Explanation

Introduction

General observation: It is important to highlight what other researchers have not done, highlight the gap, and emphasize what will contribute to this field of knowledge.

Thank you for your valuable feedback. We completely agree with your suggestion to better highlight what other researchers have not addressed and to emphasize how our study contributes to the existing body of knowledge. In response, we have revised the Introduction to more clearly identify the gap in the literature regarding the comparative effects of Light mode and Dark mode on eye strain, dryness, and visual fatigue.

We specifically pointed out that while previous research has focused on general symptoms of eye strain or isolated factors like brightness, there has been limited exploration of the immediate effects of different screen modes on visual discomfort and eye health. This gap in the literature is what our study aims to fill. By systematically comparing Light mode and Dark mode, our research provides new insights that could influence recommendations for optimal tablet usage, particularly for those engaging in prolonged screen time.

Furthermore, we have emphasized how our study will contribute to the field by offering a clearer understanding of how different display settings may impact user comfort, which can be valuable for both consumers and technology developers looking to enhance device usability and mitigate eye strain.

We believe these revisions better highlight the importance of our research and its potential to make a meaningful contribution to the field of visual health and technology design.

Improvements

-       Line 40: advantage or disadvantage?

-       You started a good paragraph using statistics to generalize the research problem. However, you did not put citations to statements in lines 41 and 43: “Studies indicate that the prevalence of Computer Vision Syndrome (CVS) among computer users in Thailand is 88%, while a study in the United States found that 75% of users experienced eye strain.” Additionally, research from “Japan showed a 72.1% prevalence rate for eye pain, a common symptom of eye strain. This research collectively points to a significant issue: most tablet users experience eye fatigue correlated with gender and age.” Two things: first of all, move from this paragraph the synthesis of research between lines 46-53 to the paragraph of research synthesis and complete the generalized paragraph with some additional statistics.

Thank you for your valuable feedback. We have revised the wording in line 40 as suggested. The sentence has been updated to improve clarity and flow.

Thank you for your valuable feedback. In response to your comments, we have made the following revisions:

  • We have added the necessary citations to support the statements in lines 41 and 43 regarding Computer Vision Syndrome (CVS) has been reported in various countries.
  • We have moved the synthesis of research between lines 46-53 to the appropriate section of the manuscript dedicated to research synthesis, as suggested.

We hope these revisions adequately address your concerns. Thank you once again for your constructive feedback.

You should reorganize the paragraphs from lines 54 to 73. So, I recommend creating a paragraph to synthesize previous research related to the effect of eye fatigue associated with gender and condense the other previous research related to prolonged exposure to the tablet screen in another paragraph. Additionally, it presents the gap or gap in existing knowledge, justified with the syntheses of previous research. It is necessary to aggregate more references in every paragraph to support the research gap.

Thank you for your constructive feedback. We have made the necessary revisions in the introduction as suggested. The paragraphs have been reorganized to synthesize the previous research more effectively, with a clearer focus on the effect of eye fatigue related to gender and prolonged exposure to tablet screens. Additionally, we have condensed the relevant research into more cohesive paragraphs and highlighted the existing research gaps to better justify the need for our study.

We appreciate your valuable input, and we believe these changes have improved the clarity and flow of the introduction.

Materials and methods

Instruments: I recommend organizing the equipment and its function in the experiment with an index. It should be clarified whether frequency measurement equipment was used to generate a normalization before testing with the tablets (despite being clarified in line 141). Additionally, the Snellen Chart, a workstation with standard measurements, an adjustable chair, a screen with adjustable brightness, and a post-test must be listed as part of the materials, equipment, and instruments (Is this equipment different from tablets?).

Thank you for your valuable feedback. We have revised the manuscript according to your recommendations and believe that the changes made have improved the clarity and comprehensiveness of the methodology section. Please let us know if any further adjustments are needed.

Procedures: in lines 136 and 137, you used “would” in the sentence, “After that, they would take a break for 30 minutes, or until they no longer experienced eye strain” and “Following the break, participants would then play games and read on the tablet screen in dark mode.” Could you improve the sentence?

Thank you for your feedback. We have revised the sentences as follows to improve clarity:

"Afterwards, participants would take a 30-minute break, or they could extend the break until they no longer experienced any symptoms of eye strain, such as discomfort or dryness. Following the break, they would resume activities, including playing games and reading, using the tablet screen in dark mode to assess its impact on eye strain and visual comfort."

We hope this revision addresses your concern by providing more detailed and clear information about the procedures. Please let us know if further modifications are required.

Results

Demographic Characteristics: Table 1 is unnecessary; the previous paragraph presented the information.

Thank you for your helpful feedback. We agree with your observation that the demographic characteristics presented in Table 1 are already described in the previous paragraph. In response, we have removed Table 1 from the manuscript to avoid redundancy. The relevant demographic information is now solely included in the narrative, streamlining the presentation of the data.

We appreciate your suggestion, which has helped improve the clarity and conciseness of the manuscript.

Tables 2, 3, and 4 missed the pre-p-values.

Thank you for pointing this out. We apologize for the oversight. We have updated all Tables to include the missing pre-p-values as requested. This correction ensures that all relevant statistical information is properly presented and enhances the clarity of the results.

We appreciate your attention to detail, which has helped improve the accuracy and completeness of the manuscript.

Discussion

Section 5.1: there are statements in the text of paragraphs 1 and 2 that require support with quotes. In addition, in paragraph 3, which text in line 245: “Previous studies comparing the effects of light mode and dark mode on visual fatigue in users of virtual reality (VR) glasses found that Dark mode significantly reduced visual fatigue compared to Light mode.”

Thank you for your feedback. We have made the necessary revisions to Section 5.1. Specifically, we have added supporting references and quotes to the statements. Additionally, we have clarified and appropriately cited the previous studies. We believe these changes address your concerns and improve the overall quality of the section.

In general, statements in the discussion should be supported with citations.

Thank you for your valuable feedback. We agree that statements in the discussion should be supported with appropriate citations. We have revised the discussion section to include relevant citations where necessary to support our statements. These additions help strengthen the validity of the arguments and ensure proper referencing of previous research.

Conclusion

Please do not present more about the results in the conclusion. Summarize your thoughts and convey the broader importance of your study. Identify how a gap in the literature has been addressed. Explain why the results are relevant. Finally, introduce future work derived from the research.

Thank you for your insightful comment. We have revised the conclusion to align with your suggestion by focusing on summarizing the broader implications of the study rather than reiterating the results. The revised conclusion now emphasizes the importance of our findings in addressing a gap in the literature, particularly in understanding how different screen modes (light mode and dark mode) impact visual fatigue and eye health during prolonged tablet use.

We also clarify the relevance of the results in offering practical recommendations for users and highlighting potential improvements for digital device design. In response to your suggestion, we have also introduced future directions for research, such as exploring the long-term effects of screen modes and the influence of environmental factors on visual discomfort.

We believe these revisions strengthen the conclusion by focusing on the broader significance of the study and its potential for further research in the field.

Limitations

Please do not put limitations on future work.

Thank you for your valuable feedback. We appreciate your suggestion to avoid placing limitations on future research. In response to your comment, we have revised the section on "Future Work" to present broader avenues for future studies, without narrowing the scope.

Specifically, we have reworded the proposed future studies to emphasize a wider range of possibilities and to allow for further exploration in various contexts. For example, we expanded the scope of potential research populations and methodologies, ensuring that the future work is framed in a way that invites further investigation without limitation.

We believe these changes align more closely with your suggestion and better reflect the potential for future research in this area.

Thank you again for your insightful input.

References

The references are between the periods of 2020 and 2023. Although, some are found between 2015 and 2019. It is necessary to increase the number of references (this must be corrected when making the proposed citation improvements)

Thank you for your valuable feedback.

We have increased the number of references as suggested, ensuring a broader and more diverse range of references to align with your recommendation.

Reviewer 3 Report

Comments and Suggestions for Authors

The following sentence, which begins at line 247 is incorrect:  "This was attributed to the reduction in pupil dilation in dark mode, which enhanced the clarity of the retinal image and decreased the occurrence of spherical aberration."

The pupils are more dilated in dark mode, which decreases image clarity and increases spherical aberration.  This context is correctly applied in reference 13 and the authors appear to have misinterpreted it in their statement at line 247.

Author Response

Reviewer 3: (Green highlight)

Comments

Editing/ Explanation

Overall

The following sentence, which begins at line 247 is incorrect:  "This was attributed to the reduction in pupil dilation in dark mode, which enhanced the clarity of the retinal image and decreased the occurrence of spherical aberration."

The pupils are more dilated in dark mode, which decreases image clarity and increases spherical aberration.  This context is correctly applied in reference 13 and the authors appear to have misinterpreted it in their statement at line 247.

Thank you very much for your valuable feedback. We acknowledge the error in the statement regarding pupil dilation in dark mode. We have revised the sentence in line 247 as follows:

"The dilation of the pupil in low-light conditions, such as Dark mode, allows more light to enter the eye but also leads to increased aberrations, thereby reducing image sharpness and enhancing spherical aberration."

This is consistent with the findings presented in reference 21, which we had inadvertently misinterpreted in our original statement.

We sincerely appreciate your attention to this detail, and we have made the necessary corrections in the manuscript. Thank you once again for your constructive input.

Reviewer 4 Report

Comments and Suggestions for Authors

The manuscript deals with the topic of the impact on visual fatigue of using different modes (dark or light polarity) for tablet users.
The authors' study is interesting and deserves to be published, but the manuscript needs some changes.
1) The authors seem to have overlooked very important studies done by other research groups. The study should be better placed in the framework of the current literature considering, as example https://doi.org/10.3233/WOR-162370, https://doi.org/10.1016/S0169-8141(99)00040-2, https://doi.org/10.1016/j.displa.2008.12.001. Some comments/comparisons with the results presented in the above-mentioned works should be made.
2) Are the lighting conditions Natural, artificial, mixed ? … this is not very clear.
3) Were there any illuminance measurements on the surface where the tablet is placed and on the user's eye ?
4) The measurement set-up should be well described, possibly also adding a schematic figure.
5) Some considerations should be added by the authors, to clarify which are the variables not controlled in their study and that the results they found are attributable only to the variables they controlled, otherwise it is difficult to trust the results obtained.
6) What are the limitations of this study? a section (or a paragraph in the conclusions) should be added to specify them.

Author Response

Reviewer 4: (Pink highlight)

Comments

Editing/ Explanation

Introduction

The authors seem to have overlooked very important studies done by other research groups. The study should be better placed in the framework of the current literature considering, as example https://doi.org/10.3233/WOR-162370, https://doi.org/10.1016/S0169-8141(99)00040-2, https://doi.org/10.1016/j.displa.2008.12.001. Some comments/comparisons with the results presented in the above-mentioned works should be made.

Thank you for your valuable feedback. We apologize for overlooking the important studies mentioned. We have now incorporated the recommended studies into the literature review section of the manuscript, and the necessary revisions have been made to better align our study with the current literature and to support our research findings.

Materials and methods

Are the lighting conditions Natural, artificial, mixed ? … this is not very clear.

Thank you for your comment. We apologize for the lack of clarity regarding the lighting conditions. The lighting conditions in the experiment were artificial. Specifically, the environment was set up with artificial lighting to ensure consistent and controlled illumination throughout the test. We hope this clears up the confusion.

Were there any illuminance measurements on the surface where the tablet is placed and on the user's eye ?

Thank you very much for your valuable feedback. The lighting conditions in the experiment were carefully controlled throughout the test. The room's lighting was maintained at a level between 300-500 lux, which is considered optimal for this type of task, and the brightness of the tablet screen was also adjusted to ensure consistent illumination. However, illuminance measurements were not taken directly at the user's eye. Future studies could consider incorporating measurements at the user's eye level to further assess the impact of light levels on visual comfort.

The measurement set-up should be well described, possibly also adding a schematic figure.

Thank you for your valuable feedback. We have carefully revised the manuscript to provide a more detailed description of the measurement set-up. In addition, we have included the research process flowchart (Figure 1) to better illustrate the set-up and enhance the clarity of the methodology.

Some considerations should be added by the authors, to clarify which are the variables not controlled in their study and that the results they found are attributable only to the variables they controlled, otherwise it is difficult to trust the results obtained.

Thank you for your comment. We have added the necessary considerations in the limitations section of the manuscript to clarify the variables that were not controlled in our study. Additionally, we have emphasized that the results are primarily attributable to the variables we controlled. We believe these changes address your concerns and provide greater clarity regarding the study's findings.

Limitations of the study

What are the limitations of this study? a section (or a paragraph in the conclusions) should be added to specify them.

Thank you for your feedback. We would like to point out that the limitations regarding the Visual Acuity of Participants, Gender of Participants, Age Groups, and Lighting Conditions have already been addressed in Section 7, 'Limitations of the Study.' However, we will review this section to ensure that these details are clearly highlighted and provide sufficient context for the readers. If needed, we will further elaborate on these points to improve clarity.

Round 2

Reviewer 2 Report

Comments and Suggestions for Authors

All the requirements were adequately addressed. Thank you for your attention.

Reviewer 4 Report

Comments and Suggestions for Authors

The manuscript has been strongly improved by the authors.

The critical issues, highlighted by the reviewers on the text and method, have been suitably treated.

In my opinion the manuscript can be accepted.